# An Analysis of the Factors Affecting Hyporheic Exchange based on Numerical Modeling

**Jie Ren** [1,*] , **Xiuping Wang** [1], **Yinjun Zhou** [2,*], **Bo Chen** [1] **and Lili Men** [1]

[1]  State Key Laboratory of Eco−hydraulics in Northwest Arid Region of China, Xi'an University of Technology, Xi'an 710048, China; wxpyl1992@163.com (X.W.); chenbo7289@163.com (B.C.); li15029069869@163.com (L.M.)

[2]  Changjiang River Scientific Research Institute, Wuhan 430010, China

\*  Correspondence: renjie@xaut.edu.cn (J.R.); zhouyinjun1114@126.com (Y.Z.)

**Abstract:** The hyporheic zone is a transition zone for the exchange of matter and energy between surface water and subsurface water. The study of trends and sensitivities of bed hyporheic exchanges to the various influencing factors is of great significance. The surface−groundwater flow process was simulated using a multiphysics computational fluid dynamics (CFD) method and compared to previous flume experiments. Based on that, the single-factor effects of flow velocity ($u$), water depth ($H$), dune wave height ($h$), and bed substrate permeability ($\kappa$) on hyporheic exchange in the bed hyporheic zone were investigated. The sensitivity analysis of various factors ($H$, $u$, dune wavelength ($L$), $h$, bed substrate porosity ($\theta$), $\kappa$, and the diffusion coefficient of solute molecules ($D_m$)) in the surface−subsurface water coupling model was done using orthogonal tests. The results indicated that $u$, $h$, and $\kappa$ were positively related, whereas $H$ was negatively related to hyporheic exchange. $H$ and $u$ showed large effects, whereas $\kappa$, $D_m$, and $\theta$ had moderate effects, and $L$ and $h$ showed small effects on hyporheic exchange. This study provides valuable references for the protection and recovery of river ecology.

**Keywords:** hyporheic exchange; surface−groundwater flow process; influencing factors; orthogonal tests; sensitivity analysis

## 1. Introduction

The hyporheic zone plays an important role as an interface between subsurface and surface water [1,2], and has important influences on exchanges of water, nutrients and heat [3–5], pollutant migration [6,7], and quality of surface and subsurface water [8–10]. In regards to water quality, hyporheic exchange controls the temperature pattern [11,12], induces diffusion of solutes on the bed [13], increases the residence time of solutes [14], accelerates circulation of nutrients [15,16], and increases opportunities for biological and geochemical processing by decelerating migrations of dissolved and suspended substances [8,17]. Hence, hyporheic exchange has a significant effect on rivers and subsurface water systems [18,19].

Hyporheic exchanges exhibit complicated temporal and spatial variations due to influences of water fluctuations [20], in-stream structures and channel flow rates [21–23], bedform morphology [24–26], sediment penetrability [22,27–29], and rainfall patterns [20]. Recently, hyporheic exchanges have been the focus of intensive research [14,30–36]. For instance, hyporheic exchanges were measured in streams using ion or dye tracers [37]. The dynamics of the coupled system for unidirectional flow in the water column and a triangular interface on dunes were investigated [38]. Based on that, sensitivity analysis was performed via multiple computational fluid dynamics (CFD) simulations and interactions between turbulent water-column flows, current topography-driven flows

in underlying permeable sediments, and ambient subsurface water discharge from deep subsurface waters were investigated [39]. Sawyer et al. [40] proposed equations for bed pressure profiles and hyporheic exchange rates in the vicinity of a channel-spanning log that can be used to evaluate the impacts of the removal or introduction of large woody debris on hyporheic mixing. Schmadel et al. [41] developed a framework relating diel hydrologic fluctuations to hyporheic exchange in simple bedform morphologies and simulated subsurface water flows under time-varying boundary conditions using an aquifer bounded by a straight stream and hill slope. The lattice Boltzmann method has been used successfully to study some complex flows [42–45]. The lattice Boltzmann method was used by Peng et al. [43] to investigate solute transport in shallow water flows. This suggests that the present model has great potential to predict morphological change in shallow water flows.

Laboratory flume tests have also been employed to investigate hyporheic exchanges [46–49]. Tonina and Buffington [50] reported a series of recirculating flume tests to investigate hyporheic exchanges in pool-riffle channels spanning a broad range of discharge and bedform morphologies. Wu and Hunkeler [51] investigated flow processes in a conduit−sediment system using both a model resembling a siphon and a numerical model. Packman et al. [6] studied solute exchange in flat and dune-shaped beds using laboratory flume tests, where dye injections indicated that a combination of convective pore water flow and turbulent diffusion near the stream–subsurface interface is responsible for solute exchange on flat beds. Sawyer et al. [52] used laboratory flume tests and numerical simulations to quantify hyporheic fluids and heat exchanges induced by current interactions with channel-spanning logs. Fox et al. [53] used a novel laboratory flume system to investigate the effects of losing and gaining flow conditions on hyporheic exchange fluxes in a sandy streambed. Lu et al. [54] constructed a two-dimensional (2D) sand tank to study the influence of a clay lens with low permeability on the hyporheic zone under different surface flow conditions. Despite the field of hyporheic zone research being fairly mature, few studies have investigated the effects of individual factors on hyporheic exchange.

In this study, sensitivity analysis was applied to factors in hyporheic exchange via orthogonal tests and hyporheic exchange was studied by flume tests [55]. The flume used had a length, width, and cross-sectional length of 2 m, 0.3 m, and 1.5 m, respectively. Seven identical triangular ripples with a wave height of 0.02 m, wavelength of 0.2 m, and distance from trough to crest of 15 cm were designed for the flume. A surface−subsurface flow coupling model was established and compared to previous flume experiments [55]. The effects of various factors, including water depth ($H$), flow velocity ($u$), dune wave height ($h$), bed substrate porosity ($\theta$), bed substrate permeability ($k$), and coefficient of solute molecules ($D_m$) on hyporheic exchanges were investigated using orthogonal tests and sensitivity analysis. Additionally, numerical simulations of the injection of dye to the dune were developed for the surface−subsurface hyporheic exchange. The evaluation parameter selected was the time to equilibrium for solute concentrations at observation points.

## 2. Methodology

### 2.1. Governing Equations for Fluid Flow

The turbulent flow in the test section of the flume was simulated following the multiphysics computational fluid dynamic (CFD) approach of Cardenas and Wilson [38]. Turbulent flow is simulated by solving the Reynolds-averaged Navier−Stokes (RANS) equations with the $k−\omega$ turbulence closure model by Wilcox [56]. The pore water flow is simulated by solving a steady state groundwater flow model using COMSOL Multiphysics. These two sets of equations are coupled through the pressure distribution at the sediment−water interface [31]. Janseen et al. [55] used exactly the same combination of software. For an incompressible fluid, the steady state RANS equations are defined as:

$$\frac{\partial U_i}{\partial x_i} = 0 \tag{1}$$

$$\rho U_j \frac{\partial U_i}{\partial x_j} = -\frac{\partial P}{\partial x_i} + \frac{\partial}{\partial x_j}(2\mu S_{ij} - \rho \overline{u'_j u'_i}) \tag{2}$$

where $\rho$ refers to the fluid density, $\mu$ refers to the dynamic viscosity (assumed standard for water), $U_{i\ or\ j}$ ($i, j = 1, 2$ where $i \neq j$) refers to the time-averaged velocity, $u_i'$ ($i = 1, 2$) refer to the fluctuations in the instantaneous velocity components in $x_{i\ or\ j}$ ($i, j = 1, 2$, where $i \neq j$) directions, and $P$ refers to the time-averaged pressure. The strain rate tensor ($S_{ij}$) is defined as:

$$S_{ij} = \frac{1}{2}\left(\frac{\partial U_i}{\partial x_j} + \frac{\partial U_j}{\partial x_i}\right) \tag{3}$$

The Reynolds stresses, $\tau_{ij}$, are related to the mean strain rates by:

$$\tau_{ij} = -\overline{u_j' u_i'} = v_t(2S_{ij}) - \frac{2}{3}\delta_{ij}k \tag{4}$$

where $v_t$ refers to the kinematic eddy viscosity, $\delta_{ij}$ refers to the Kronecker delta, and $k$ refers to the turbulent kinetic energy. The $k-\omega$ turbulence closure scheme [56] was employed due to its advantages in simulating separated flows with adverse pressure gradients, including flow over dunes where a pronounced eddy is present [57,58]. The eddy viscosity in this closure scheme is:

$$v_t = \frac{k}{\omega} \tag{5}$$

where the specific dissipation, $\omega$, is defined as the ratio of the turbulence dissipation rate $\varepsilon$ to $k$:

$$\omega = \frac{\varepsilon}{\beta^* k} \tag{6}$$

where $\beta^*$ refers to the closure coefficient.

The steady-state migration equations for $k$ and $\omega$ are:

$$\rho\frac{\partial(U_j k)}{\partial x_j} = \rho\tau_{ij}\frac{\partial U_i}{\partial x_j} - \beta^*\rho\omega k + \frac{\partial}{\partial x_j}\left[(\mu + \mu_t\sigma_k)\frac{\partial k}{\partial x_j}\right] \tag{7}$$

$$\rho\frac{\partial(U_j \omega)}{\partial x_j} = \alpha\frac{\rho\omega}{k}\tau_{ij}\frac{\partial U_i}{\partial x_j} - \beta\rho\omega^2 + \frac{\partial}{\partial x_j}\left[(\mu + \mu_t\sigma_\omega)\frac{\partial \omega}{\partial x_j}\right] \tag{8}$$

The standard closure coefficients for the $k-\omega$ scheme are from Wilcox [56]: $\alpha = 5/9$, $\beta = 3/40$, $\beta^* = 9/100$, $\sigma_k = \sigma_\omega = 0.5$.

The 2D pore water flow in sand was modeled by solving the steady-state subsurface water flow equation:

$$\nabla(-\frac{\kappa}{\mu}\nabla \times P) = 0 \tag{9}$$

where $\kappa$ refer to isotropic permeability. The parenthetical term is the Darcy flux or Darcy velocity ($Q$). The solution for simulating turbulent flow for pressure along the interface of sediment and water is described as a Dirichlet boundary at the top of the porous domain, whereas simulations of other porous flows corresponding to the flume have no flow boundaries. The flow field divided by porosity was used as the input for particle tracking, whereas dispersion is neglected as this study focuses on the convective flow paths. The pore water flows were simulated using the finite element code COMSOL Multiphysics.

The solute migration model was established based on convective diffusion equations [59]:

$$\frac{\partial C}{\partial t} = D_m\frac{\partial^2 C}{\partial x_i^2} - u_i\frac{\partial C}{\partial x_i} + \frac{\partial}{\partial x_i}\left(D_{ij}\frac{\partial C}{\partial x_i}\right) \tag{10}$$

where $C$ is solute concentration, $t$ is time, $D_m$ is the molecular diffusion coefficient in porous media, and $D_{ij}$ is the mechanical dispersion coefficient ensor. Index $i, j = 1, 2$.

The equation to obtain $D_{ij}$ is [60]:

$$D_{ij} = \alpha_T U \delta_{ij} + (\alpha_L - \alpha_T)\frac{u_i u_j}{U} \tag{11}$$

where $\alpha_T$ and $\alpha_L$ are transverse and longitudinal dispersivities, $U$ is the pore velocity magnitude, and $\delta_{ij}$ is the Kronecker delta function.

The Millington−Quirk model of the effective diffusion coefficient is:

$$\tau = \theta^{-\frac{1}{3}} \tag{12}$$

where $\theta$ refers to the porosity.

### 2.2. Calculation Model

The numerical calculation model proposed in the present study is based on the flume tests reported by Janssen et al. [55]. This model has a length of 1.5 m, including 0.05 m buffer segments at inlets and outlets and seven identical triangular dunes (crest curvature radius = 0.02 m, bed substrate height = 0.09 m, water depth $H$ = 0.1 m). Figure 1 shows the 2D flume model in which arrows describe the flow directions.

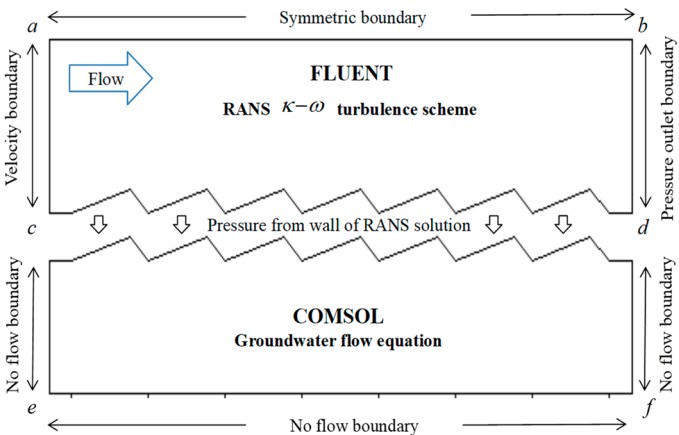

**Figure 1.** Two-dimensional flume model.

The present study assumed fluids to be steady and incompressible, bed substrates to be homogeneous, and isotropic and dune configurations to be free of any shifts. Fluent is a CFD software used for simulating and analyzing fluid flow and heat exchange problems in complex geometric regions. Using Fluent software to study the eddy current structure, flow separation, and water−sand interface pressure distribution results of surface water modeling under riverbed morphological disturbance has high precision. As shown in Figure 1, the overlying water was simulated by the CFD−Fluent; COMSOL Multiphysics is a software based on the finite element method. It is one of the softwares used for multiphysics coupling simulation by solving partial differential equations. It has good post-processing function. It is more accurate to use COMSOL software to describe the hyporheic exchange in a groundwater model. Therefore, we used the commercial finite element software COMSOL Multiphysics to solve the groundwater flow equations in the porous sediment. Pressure distributions at the interface of water and sand as determined by the CFD−Fluent were based on the coupling of surface and subsurface water.

The 6th dune was located in the middle of the flume, and the flow interference was lower. In addition, Janssen et al. [55] had measured the vertical velocity of the section above the water−sand

interface of the 5th, 6th, and 7th sections. Based on this, the 6th dune was selected as the subject. With the height of the dune trough defined as "0", four feature observation points (A, B, C, and D) at −2 cm were selected, as shown in Figure 2. More specifically, observation points A and D were situated at the trough, observation point B was situated 8 cm to the right of observation point A (1/2 of the upstream face), and observation point C was situated at the crest. The dye injection into bed substrate were simulated as circular areas with observation points as the centers and radii of 0.5 cm to monitor the hyporheic exchange routes. In the present study, $CaCl_2$ solution was used as the dye and the time to equilibrium for concentrations at observation points was chosen as the evaluation parameter; the concentration variation was reflected by the dimensionless $C/C_0$ ($C_0$ = initial concentration).

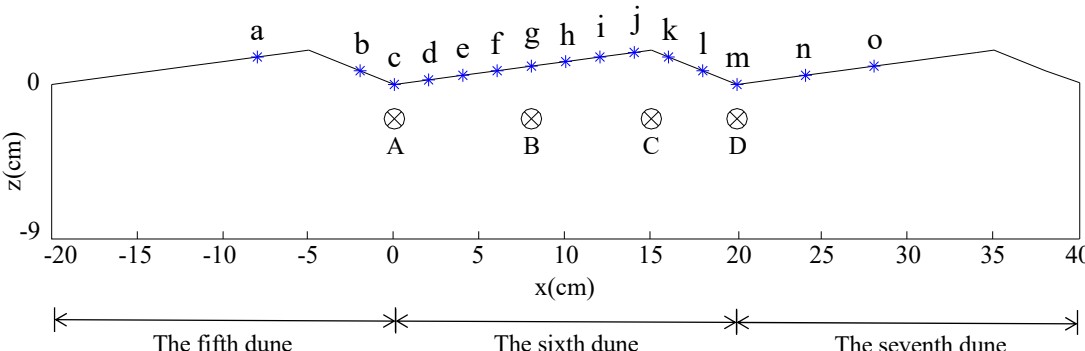

**Figure 2.** Layout of monitoring points.

Table 1 summarizes the parameters used in the proposed calculation model [55].

**Table 1.** Parameters used in the calculation model.

| Fluid Density $\rho$ (kg m$^{-3}$) | Kinematic Eddy Viscosity $\nu_t$ (m$^2$ s$^{-1}$) | Porosity $\theta$ (%) | Permeability $\kappa$ (m$^2$) | Molecular Diffusion Coefficient $D_m$ (m$^2$ s$^{-1}$) |
|---|---|---|---|---|
| 998.8 | $1.10 \times 10^{-6}$ | 40 | $1.50 \times 10^{-11}$ | $5.00 \times 10^{-11}$ |

### 2.3. Grid Division and Boundary Conditions

The overlying water grid was generated using the Gambit. The boundary layer had a thickness of 1 cm, meshes near wall boundaries had sizes of 0.2 mm, the growth rate was 1.06, and a total of 25 layers were generated. The boundary layer used had a thickness of 1.1 cm, and 440,000 meshes were generated for the overlying water. The bed substrate was divided into 10,000 meshes with the mass of 0.95. The constant velocity inlet boundary is referred to as *ac*, *bd* refers to the constant pressure outlet boundary, and *ab* refers to the free fluid surface as a symmetric boundary. In shallow water applications the top of the *ab* domain is actually a free surface, but in our simulations the water depth was large enough to replace the free surface with the symmetry condition [55]. The variable *cd* refers to the interface of water and sand as a sliding-free wall boundary. Although recent studies have emphasized the effects of wall permeability on the mass and momentum exchange at the interface of water and sand, it is a valid approximation for interfacial flow with lower porosity and permeability porous media such as considered in this study [31]. Flux free boundaries are referred to as *ce*, *ef*, and *df*.

### 2.4. Model Evaluation

Root mean square error (RMSE), coefficient of determination ($R^2$), and the relative error (Re) were used to quantitatively evaluate the accuracy of the calculation model:

$$\text{RMSE} = \sqrt{\sum_{i=1}^{n} (O_i - S_i)^2 / n} \tag{13}$$

$$\mathrm{R}^2 = 1 - \sum_{i=1}^{n} (O_i - S_i)^2 / \sum_{i=1}^{n} (O_i - \overline{O})^2 \tag{14}$$

$$\mathrm{Re} = \sqrt{\sum_{i=1}^{n} (O_i - S_i)^2 / \sum_{i=1}^{n} O_i^2} \tag{15}$$

where $O_i$, $S_i$, $n$, and $\overline{O}$ refer to the measured value, simulated value, sample size, and average value, respectively.

The consistency between measured and simulated values was measured using RMSE to verify the model. The RMSE is a non-negative value and a low RMSE indicates good consistency between measured and simulated values [61]. $\mathrm{R}^2$ is the coefficient of determination of the linear regression equation (y = x) between measured and simulated values and a large $\mathrm{R}^2$ indicates good consistency between measured and simulated values [62]. Re is the relative error between measured and simulated values and a low Re indicates good consistency between measured and simulated values [63].

*2.5. Orthogonal Tests*

Orthogonal testing involves the use of orthogonal list-based multi-factor tests and result analysis. Representative points with a homogeneous distribution and good comparability were selected for testing based on the principle of orthogonality. The orthogonal test is used to assess result trends with reduced testing cycles. The sensitivity of each factor on the evaluation index was determined by range analysis of the results of orthogonal tests.

Suppose $M$ and $N$ refer to different influencing factors, $t$ refers to the factor level, $M_i$ refers to the $i$th ($i$ = 1, 2, ..., $t$) level of factor $M$, and $X_{ij}$ refers to the $i$th ($I$ = 1, 2, ..., $n$) level of factor $j$ ($j$ = $M$, $N$). $F_s$ ($s$ = 1, 2, ..., $n$) refers to the testing result at $X_{ij}$. The statistical parameters were calculated by:

$$K_{ij} = \frac{1}{n} \sum_{s=1}^{n} F_s - \overline{F} \tag{16}$$

where $K_{ij}$, $n$, $F_s$, and $\overline{F}$ refer to the average of factor $j$ at the $i$th level, the testing cycles of factor $j$ at the $i$th value, the value of the evaluation index in the $s$th test, and the average of the evaluation index, respectively.

The range ($R_j$) is the evaluation parameter for range analysis of factor sensitivity and can be calculated by:

$$R_j = \mathrm{Max}\{K_{1j}, K_{2j}, \cdots\} - \mathrm{Min}\{K_{1j}, K_{2j}, \cdots\} \tag{17}$$

A large $R_j$ suggests that variation of a specific factor has a significant effect on the evaluation parameter, indicating high sensitivity of hyporheic exchange to this factor and vice versa.

Based on the 2D dune-shaped surface−subsurface coupling mathematical model, $u$, $H$, $h$, $L$, $\kappa$, $\theta$, and $D_m$ were selected as the factors affecting hyporheic exchange. Three levels (−20%, average, +20%) were designed for each factor. Table 2 summarizes the calculation parameters and average level of each factor.

**Table 2.** Orthogonal tests of sensitivities of affecting factors and their levels in the model.

| Factor Level | $u$ (m s$^{-1}$) | $H$ (m) | $h$ (m) | $L$ (m) | $K$ (m$^2$) | $\Theta$ (%) | $D_m$ (m$^2$ s$^{-1}$) |
|---|---|---|---|---|---|---|---|
| 1 | 0.056 | 0.080 | 0.016 | 0.160 | $1.2 \times 10^{-11}$ | 32 | $4.0 \times 10^{-11}$ |
| 2 | 0.070 | 0.100 | 0.020 | 0.200 | $1.5 \times 10^{-11}$ | 40 | $5.0 \times 10^{-11}$ |
| 3 | 0.084 | 0.120 | 0.024 | 0.240 | $1.8 \times 10^{-11}$ | 48 | $6.0 \times 10^{-11}$ |

## 3. Results and Discussion

### 3.1. Model Validation

The model was calibrated using data obtained at monitoring points c–m on the 6th dune under low flow conditions (2.1 L s$^{-1}$) and the calibrated model was further validated by using data obtained at monitoring points a, b, n, and o on the 5th and the 7th dunes, as shown in Figure 2. The average flow velocity was 0.07 m s$^{-1}$. Vertical simulated and measured values of velocity vectors at the monitoring points were obtained by the particle image velocimetry (PIV) technique by unifying model size and boundary conditions, as shown in Figure 3. Table 3 shows the results of the evaluation of the model simulation accuracy. A close correlation between observed and measured values was found, with RMSEs at monitoring points c–m on the 6th dune ranging between 0.0025–0.0055 m s$^{-1}$, significantly lower than the average flow velocity; R$^2$ at monitoring point k was 0.8935, whereas those at the other monitoring points exceeded 0.9; Re was 3.26–7.13%. Monitoring points a and b on the 5th dune and monitoring points n and o on the 7th dune were then investigated. The RMSEs obtained were 0.0033–0.0063 m s$^{-1}$, significantly lower than the average flow velocity; the R$^2$ values of b, n, and o exceeded 0.95, whereas that of a was 0.88; Re was in a reasonable range (4.7–8.53%). In summary, the proposed model exhibited excellent simulation accuracy and can precisely describe the dynamic migration of bed solutes.

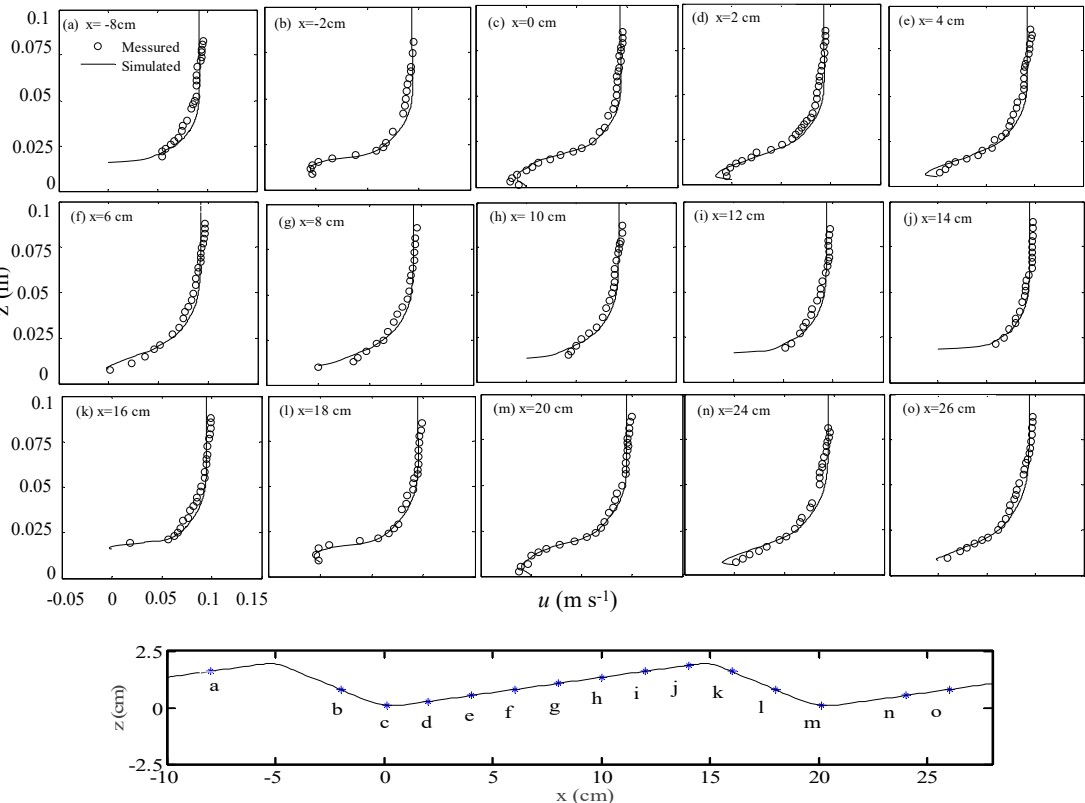

**Figure 3.** Comparison of vertical simulation results and experimental velocity vector results at different monitoring points (adapted from Janssen et al. [55]).

**Table 3.** Accuracy of vertical simulation of velocity vectors at different monitoring points.

| Monitoring Points | RMSE (m s$^{-1}$) | $R^2$ | Re % |
|:---:|:---:|:---:|:---:|
| a | 0.0046 | 0.8831 | 5.61 |
| b | 0.0033 | 0.9918 | 4.70 |
| c | 0.0044 | 0.9890 | 5.81 |
| d | 0.0047 | 0.9808 | 6.48 |
| e | 0.0054 | 0.9653 | 7.13 |
| f | 0.0055 | 0.9242 | 6.76 |
| g | 0.0050 | 0.9606 | 6.53 |
| h | 0.0053 | 0.9038 | 6.59 |
| i | 0.0044 | 0.8935 | 5.13 |
| j | 0.0034 | 0.9009 | 3.79 |
| k | 0.0042 | 0.9480 | 4.93 |
| l | 0.0041 | 0.9856 | 5.25 |
| m | 0.0025 | 0.9957 | 3.26 |
| n | 0.0063 | 0.9553 | 8.53 |
| o | 0.0039 | 0.976 | 5.05 |

*3.2. Effects of Flow Velocity on Hyporheic Exchange*

As flow velocity is a key factor affecting hyporheic exchange on a bed downstream of a dam, hyporheic exchanges at $u = 0.056$ m s$^{-1}$, 0.070 m s$^{-1}$, and 0.084 m s$^{-1}$ were investigated. Figure 4 shows the cloud chart of concentrations. Depth and range of hyporheic exchange of solutes per unit time showed a positive relationship with $u$ owing to interactions of surface water and subsurface water; seepage depth was 3.8 cm, 4.5 cm, and 5.5 cm and at $u = 0.056$ m s$^{-1}$, 0.070 m s$^{-1}$, and 0.084 m s$^{-1}$, respectively. In other words, the flow velocity was positively related to hyporheic exchange.

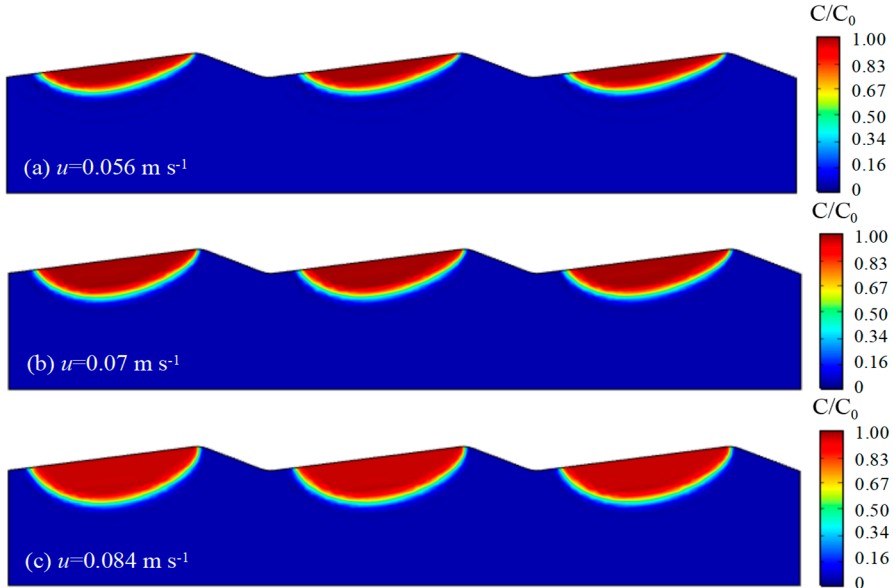

**Figure 4.** Distribution of concentration.

Table 4 shows the velocity effect on fluxes in hyporheic exchange, with flow velocity showing a positive relationship with water and solute fluxes on the interface of water and sand. In the vertical direction, opposite directions in water flux were observed between the upstream and downstream faces, demonstrating the hyporheic exchange process.

**Table 4.** Hyporheic exchange flux at different flow velocities.

| | Vertical Water Flux on the Upstream Face (m² s⁻¹) | Vertical Water Flux on the Downstream Face (m² s⁻¹) | Overall Water Flux at the Interface (m² s⁻¹) | Overall Solute Flux at the Interface (mol m⁻¹ s⁻¹) |
|---|---|---|---|---|
| $u = 0.056$ m s$^{-1}$ | $-1.42 \times 10^{-8}$ | $5.20 \times 10^{-9}$ | $8.12 \times 10^{-8}$ | $2.49 \times 10^{-7}$ |
| $u = 0.070$ m s$^{-1}$ | $-2.25 \times 10^{-8}$ | $8.28 \times 10^{-9}$ | $1.29 \times 10^{-7}$ | $4.26 \times 10^{-7}$ |
| $u = 0.084$ m s$^{-1}$ | $-3.28 \times 10^{-8}$ | $1.21 \times 10^{-8}$ | $1.88 \times 10^{-7}$ | $6.51 \times 10^{-7}$ |

Figure 5 shows stress distributions at $u = 0.056$ m s$^{-1}$, 0.070 m s$^{-1}$, and 0.084 m s$^{-1}$. Stress distributions were consistent across different flow velocities. The maximum, minimum, and negative pressure zones were observed at midstream of the upstream face, at the crest, and the downstream face, respectively. In addition, the pressure increased continuously with $u$.

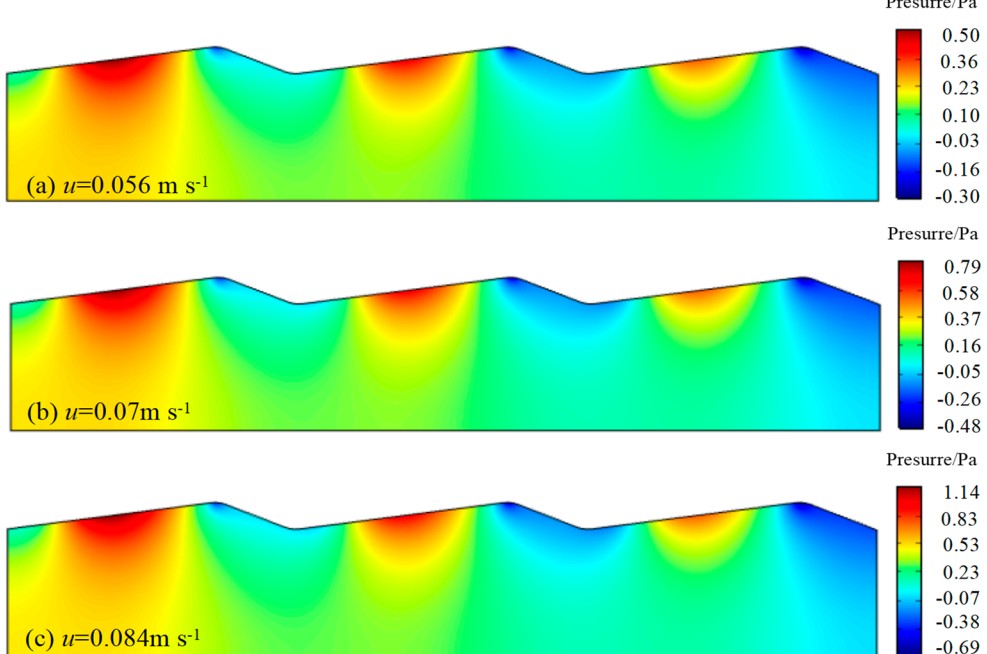

**Figure 5.** Distribution of the pressure field.

Figure 6 shows concentrations at the observation points over time. The curve gradient reflects the hyporheic exchange rate. As shown in Figure 6, $u$ had negative and positive relationships with the time to equilibrium for concentrations at observation points ($t$) and the curve gradient, respectively, indicating an increasing hyporheic exchange rate. A minimum $t$ of observation point B was evident at the upstream slope, indicating a maximum hyporheic exchange rate at this point. This can be attributed to the fact that maximum pressure was observed at midstream of the upstream face, which has a greater impact on observation point B.

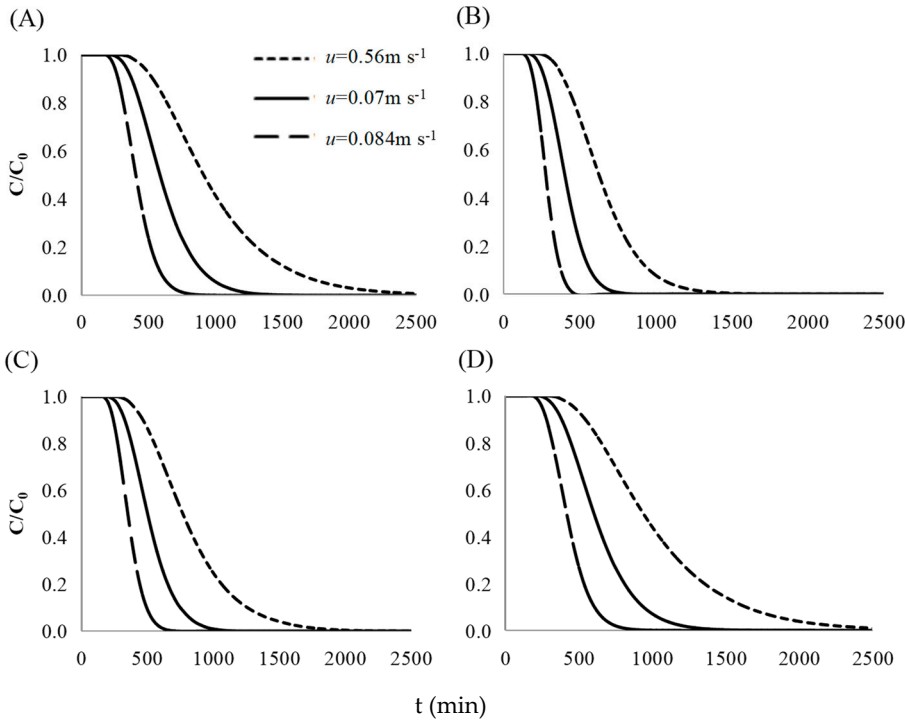

**Figure 6.** Concentrations at different monitoring points over time: (**A**) Observation point A, (**B**) Observation point B, (**C**) Observation point C, (**D**) observation point D.

### 3.3. Effects of Water Depth on Hyporheic Exchange

Hyporheic exchanges at $H = 0.08$ m, 0.1 m, and 0.12 m were investigated as $H$ has a significant effect on hyporheic exchange. Figure 7 shows the cloud chart of concentrations. $H$ evidently showed negative relationships with hyporheic exchange and the migration depth of solute per unit time, resulting in a reduced affected area. More specifically, the seepage depth was 5.1 cm, 4.5 cm, and 4.3 cm at $H = 0.08$ m, 0.1 m, and 0.12 m, respectively. In other words, $H$ was negatively related to hyporheic exchange.

Table 5 summarizes hyporheic exchange fluxes as a function of $H$. As observed, hyporheic exchange flux at the interface of water and sand showed a negative relationship with $H$. In the vertical direction, fluxes on the upstream and downstream faces were different and in opposite directions, indicating the presence of hyporheic exchange.

**Table 5.** Hyporheic exchange flux at different water depths.

| | Vertical Water Flux on the Upstream Face ($m^2\ s^{-1}$) | Vertical Water Flux on the Downstream Face ($m^2\ s^{-1}$) | Overall Water Flux at the Interface ($m^2\ s^{-1}$) | Overall Solute Flux at the Interface ($mol\ m^{-1}\ s^{-1}$) |
|---|---|---|---|---|
| $H = 0.08$ m | $-2.86 \times 10^{-8}$ | $1.70 \times 10^{-8}$ | $1.62 \times 10^{-7}$ | $5.50 \times 10^{-7}$ |
| $H = 0.10$ m | $-2.25 \times 10^{-8}$ | $8.28 \times 10^{-9}$ | $1.29 \times 10^{-7}$ | $4.26 \times 10^{-7}$ |
| $H = 0.12$ m | $-2.05 \times 10^{-8}$ | $7.45 \times 10^{-9}$ | $1.19 \times 10^{-7}$ | $3.86 \times 10^{-7}$ |

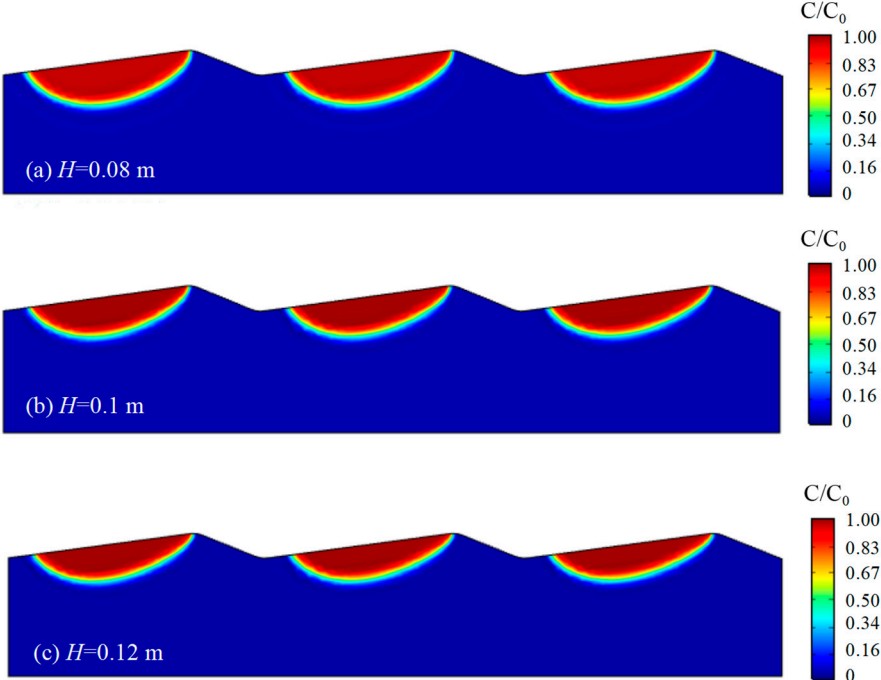

**Figure 7.** Distribution of concentrations.

Figure 8 shows that the stress distributions as a function of *H* were highly consistent with those as a function of *u*. In other words, the maximum and minimum pressures were observed at midstream of the upstream face and the crest, whereas the pressure values were different. In addition, the pressure decreased as *H* increased.

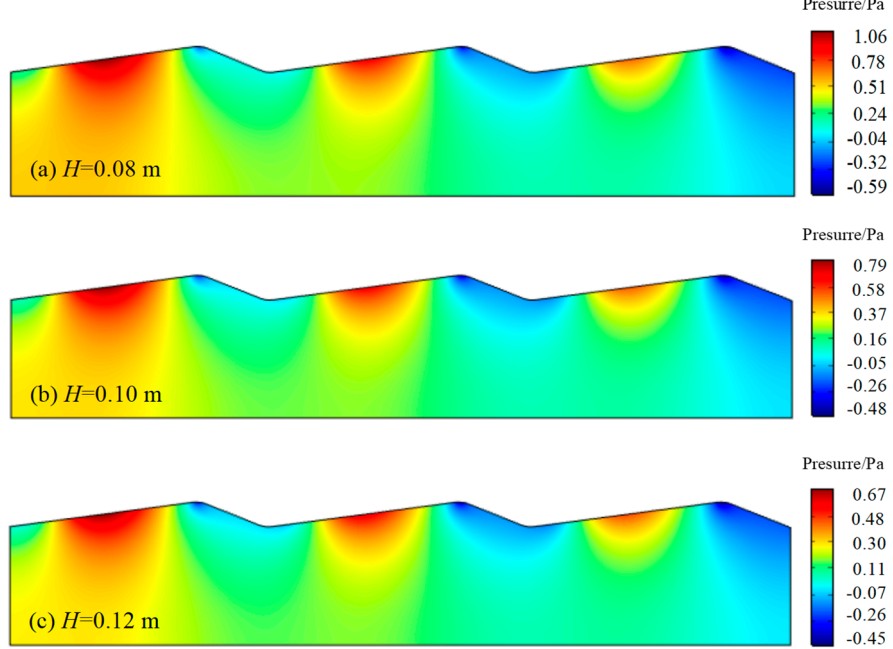

**Figure 8.** Distribution of the pressure field.

Figure 9 shows the solute concentrations at observation points A, B, C, and D over time as a function of *H*. A positive relationship between *H* and *t* was evident, indicating that hyporheic exchange was negatively related to *H*. The curve gradient, and thus the hyporheic exchange rate, was maximum

at observation point B, followed by that at observation point C, whereas the hyporheic exchange rates of observation points A and D at the trough were similar and relatively low.

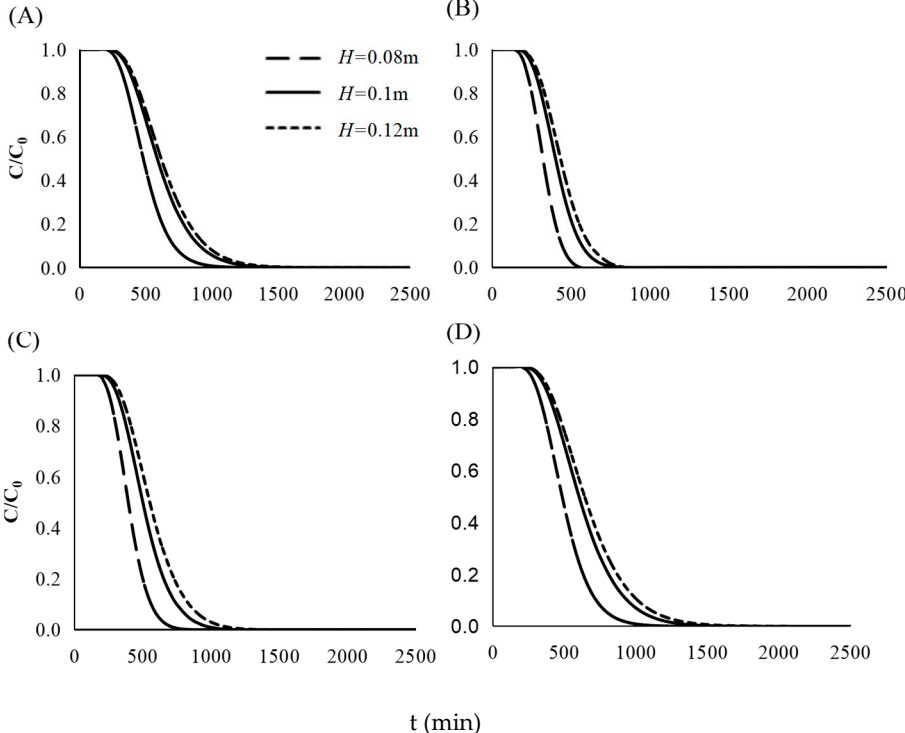

**Figure 9.** Concentrations at different monitoring points over time: (**A**) Observation point A, (**B**) Observation point B, (**C**) Observation point C, (**D**) observation point D.

## 3.4. Effects of Dune Wave Height on Hyporheic Exchange

As irregular bedform morphology has significant effects on hyporheic exchange, hyporheic exchanges at $h = 0.016$ m, 0.02 m, and 0.024 m were investigated. Figure 10 shows the cloud chart of concentrations. $h$ showed positive relationships with surface roughness, depth, and range of solutes. In other words, hyporheic exchange increased with $h$; seepage depth was 4.1 cm, 4.5 cm, and 5.2 m at $h = 0.016$ m, 0.02 m, and 0.024 cm, respectively.

Table 6 shows hyporheic exchange flux as a function of $h$, where it is evident that bedform morphology roughness showed a positive relationship with the overall water and solute fluxes in hyporheic exchange at the sediment−water interfaces.

**Table 6.** Hyporheic exchange flux at different dune wave heights.

| | Vertical Water Flux on the Upstream Face ($m^2 s^{-1}$) | Vertical Water Flux on the Downstream Face ($m^2 s^{-1}$) | Overall Water Flux at the Interface ($m^2 s^{-1}$) | Overall Solute Flux at the Interface ($mol\ m^{-1} s^{-1}$) |
|---|---|---|---|---|
| $h = 0.16$ m | $-1.62 \times 10^{-8}$ | $8.18 \times 10^{-9}$ | $1.03 \times 10^{-7}$ | $3.20 \times 10^{-7}$ |
| $h = 0.20$ m | $-2.25 \times 10^{-8}$ | $8.28 \times 10^{-9}$ | $1.29 \times 10^{-7}$ | $4.26 \times 10^{-7}$ |
| $h = 0.24$ m | $-2.93 \times 10^{-8}$ | $8.63 \times 10^{-9}$ | $1.58 \times 10^{-7}$ | $5.53 \times 10^{-7}$ |

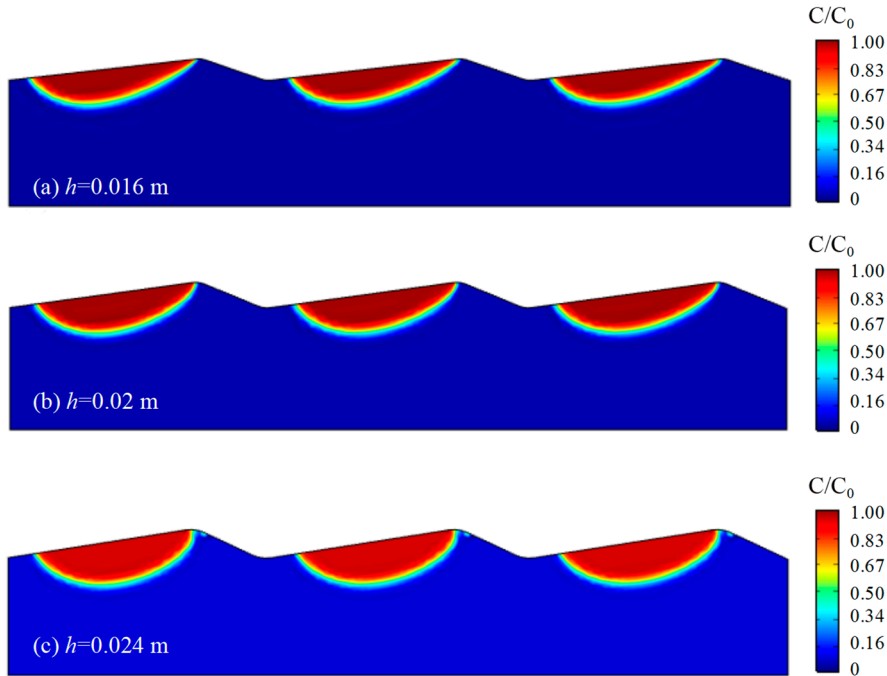

**Figure 10.** Distribution of concentration.

Figure 11 shows that the stress distributions as a function of *h* were highly consistent with those as a function of *u* and *H*, although the stress values were different. Indeed, the maximum pressure at *h* = 0.024 m was twice that at *h* = 0.016 m, whereas pressures in the negative pressure zone were homogeneous.

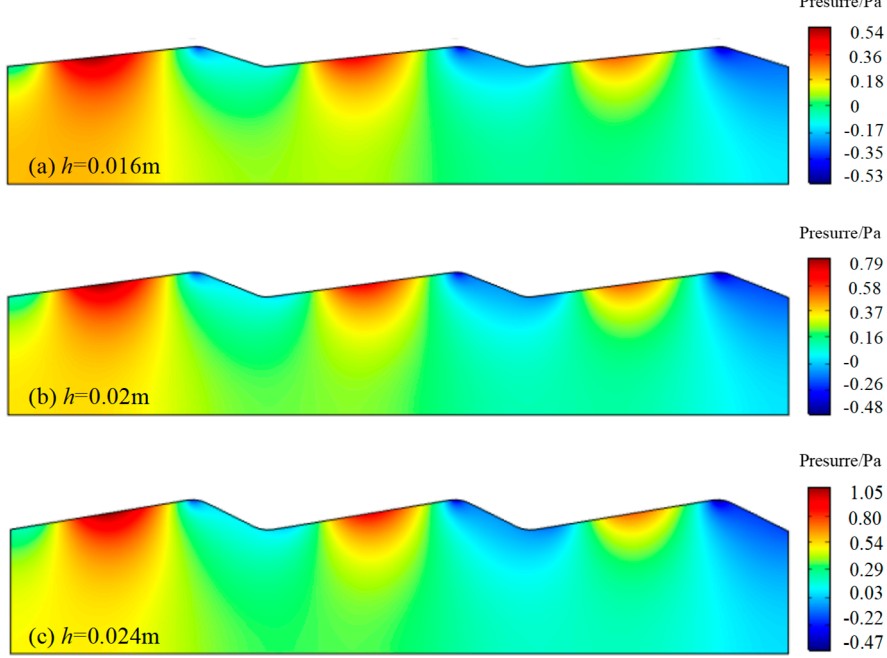

**Figure 11.** Distribution of pressure field.

Figure 12 shows the solute concentrations at observation points A, B, C, and D over time as a function of *h*. *h* showed a negative relationship with *t* at observation points A, B, C, and D, resulting in enhanced and accelerated hyporheic exchange. Hence, *h* was positively related to hyporheic exchange.

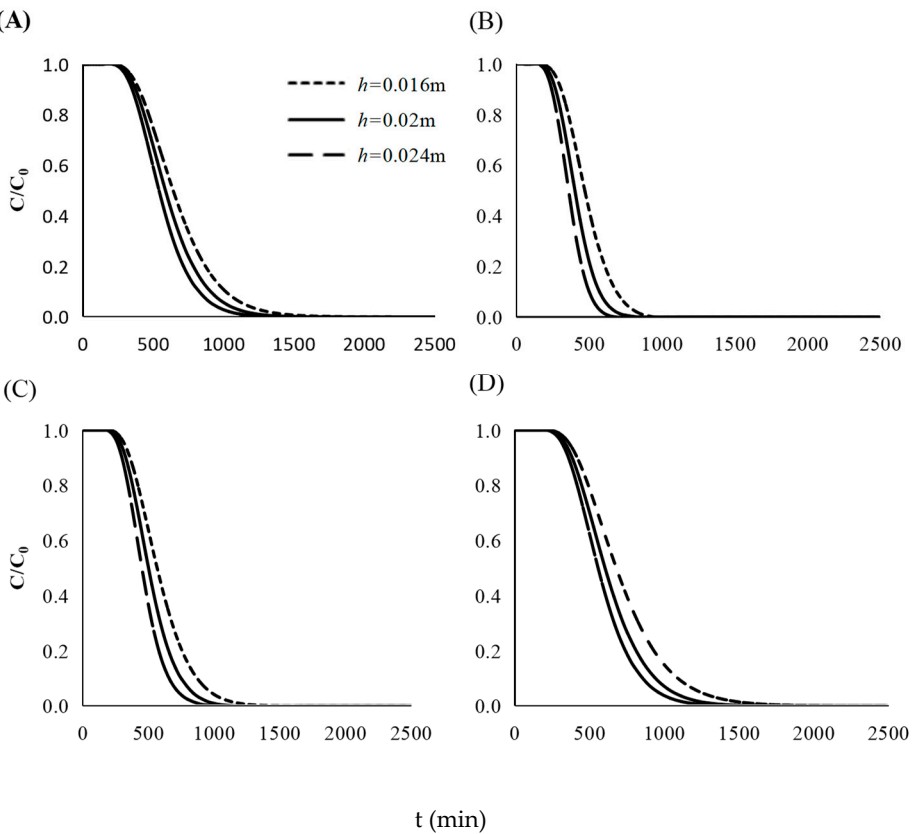

**Figure 12.** Concentrations at different monitoring points as a function of time: (**A**) Observation point A, (**B**) Observation point B, (**C**) Observation point C, (**D**) observation point D.

### 3.5. Effects of Bed Substrate Permeability on Hyporheic Exchange

As the structure of bed sediments has a significant effect on hyporheic exchange and the permeability of bed substrate has a determining effect on the hyporheic exchange depth, hyporheic exchanges at $\kappa = 1.2 \times 10^{-11}$ m$^2$, $1.5 \times 10^{-11}$ m$^2$, and $1.8 \times 10^{-11}$ m$^2$ were investigated. Figure 13 shows the cloud chart of concentrations in these cases. Seepage depth was 4.2 cm, 4.5 cm, and 5 cm at $\kappa = 1.2 \times 10^{-11}$ m$^2$, $1.5 \times 10^{-11}$ m$^2$, and $1.8 \times 10^{-11}$ m$^2$, respectively. In other words, the depth and range of solute exchange increased with $\kappa$, indicating that hyporheic exchange was positively related to $\kappa$.

Table 7 shows hyporheic exchange flux as a function of $\kappa$. Bed substrates with high permeabilities accelerated hyporheic exchange by accelerating the penetration of fluids and solutes through the interface of water and sand.

**Table 7.** Hyporheic exchange flux at different bedform permeabilities.

| | Vertical Water Flux on the Upstream Face (m$^2$ s$^{-1}$) | Vertical Water Flux on the Downstream Face (m$^2$ s$^{-1}$) | Overall Water Flux at the Interface (m$^2$ s$^{-1}$) | Overall Solute Flux at the Interface (mol m$^{-1}$ s$^{-1}$) |
|---|---|---|---|---|
| $\kappa = 1.2 \times 10^{-11}$ m$^2$ | $-1.80 \times 10^{-8}$ | $6.63 \times 10^{-9}$ | $1.03 \times 10^{-7}$ | $3.30 \times 10^{-7}$ |
| $\kappa = 1.5 \times 10^{-11}$ m$^2$ | $-2.25 \times 10^{-8}$ | $8.28 \times 10^{-9}$ | $1.29 \times 10^{-7}$ | $4.26 \times 10^{-7}$ |
| $\kappa = 1.8 \times 10^{-11}$ m$^2$ | $-2.70 \times 10^{-8}$ | $9.94 \times 10^{-9}$ | $1.55 \times 10^{-7}$ | $5.23 \times 10^{-7}$ |

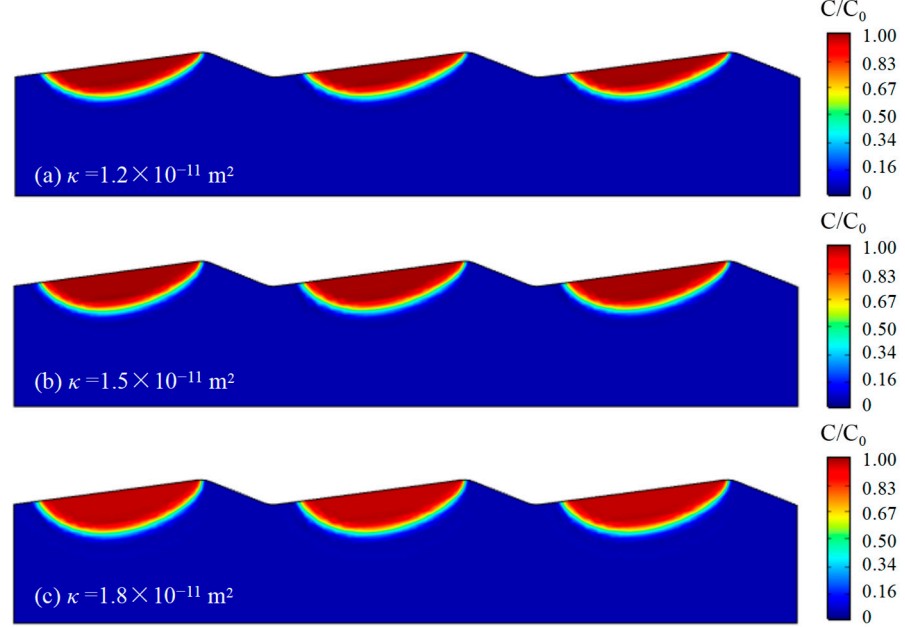

**Figure 13.** Distribution of concentration.

Figure 14 shows stress distributions as a function of $\kappa$, with no variations in stress as $u$, $H$, and $h$ remained constant. Figure 15 shows the solute concentrations at observation points A, B, C, and D over time. A negative relationship was evident between $\kappa$ and $t$, demonstrating that increasing $\kappa$ accelerates hyporheic exchange. In addition, stress distributions as a function of $\kappa$ were highly consistent with those as a function of $u$, $H$, and $h$. More specifically, the order of the curve gradient from largest to smallest was that at observation point B (maximum hyporheic exchange rate), followed by that at observation point C at the crest, and then that at observation points A and D at the trough.

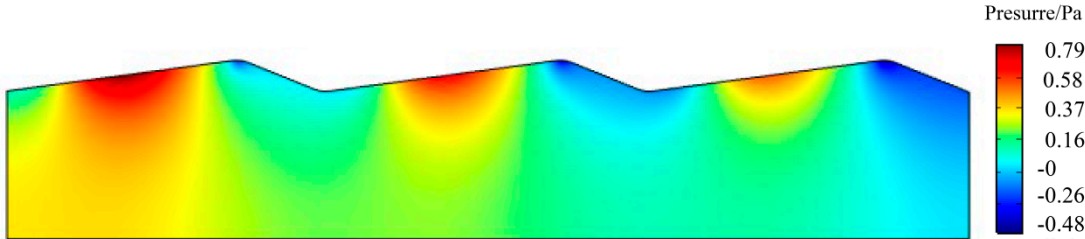

**Figure 14.** Distribution of pressure fields on bedforms with different permeabilities.

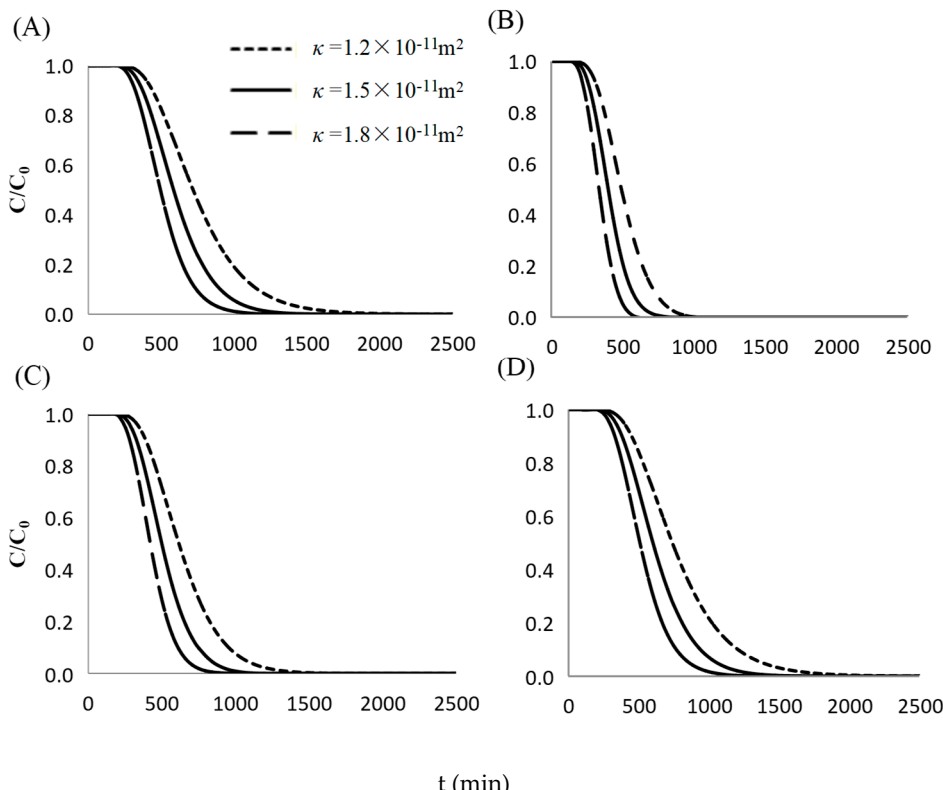

t (min)

**Figure 15.** Concentrations at different monitoring points as a function of time: (**A**) Observation point A, (**B**) Observation point B, (**C**) Observation point C, (**D**) observation point D.

### 3.6. Sensitivity Analysis

To investigate the process and intensity of hyporheic exchange, 18 orthogonal tests were designed. Observation points I and III at the trough and II at the crest on the 6th dune (see Figure 16) were selected and injections of dye ($CaCl_2$ solution in this case) into the bed substrate were simulated by circles with observation points as centers and radii of 0.5 cm to monitor the hyporheic exchange routes. $t$ at observation points I, II, and III ($t_I$, $t_{II}$, and $t_{III}$, respectively) were employed as the evaluation indices. As shown in Figure 16, the minimum, negative, and maximum pressure zones were observed at the crest, the downstream face, and trough and the upstream face, respectively.

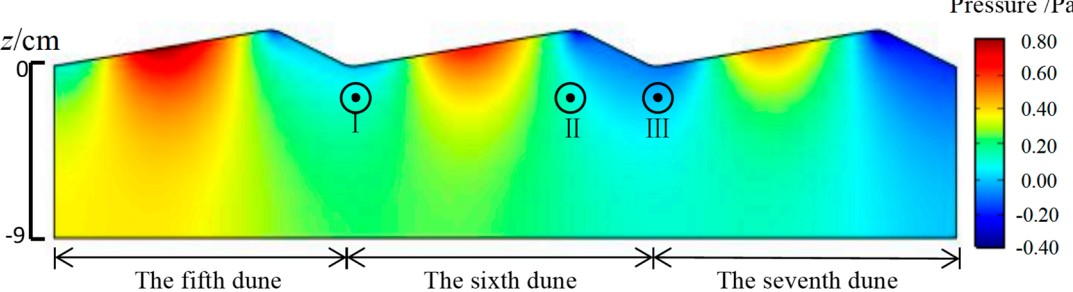

**Figure 16.** Cloud charts of pressure distributions at the 5th, 6th and 7th dunes of observation points I, II, and III.

Table 8 summarizes the schemes and results of the $L_{18}$ ($2 \times 3^7$) orthogonal tests of factors affecting hyporheic exchange. Tables 9–11 show the results of range analysis of factors affecting hyporheic exchange at observation points I, II, and III, respectively, based on $t_I$, $t_{II}$, and $t_{III}$, respectively.

**Table 8.** Schemes and results of the $L_{18}$ ($2 \times 3^7$) orthogonal tests of factors affecting hyporheic exchange.

| Test | Empty | $u$ (m s$^{-1}$) | $H$ (m) | $h$ (m) | $L$ (m) | $\kappa$ (m$^2$) | $\theta$ (%) | $D_m$ (m$^2$ s$^{-1}$) | $t_I$ (min) | $t_{II}$ (min) | $t_{III}$ (min) |
|---|---|---|---|---|---|---|---|---|---|---|---|
| 1 | 1 | 0.056 | 0.08 | 0.016 | 0.16 | $1.20 \times 10^{-11}$ | 32 | $4.00 \times 10^{-11}$ | 3183 | 2155 | 3411 |
| 2 | 1 | 0.056 | 0.1 | 0.02 | 0.2 | $1.50 \times 10^{-11}$ | 40 | $5.00 \times 10^{-11}$ | 3800 | 2776 | 4027 |
| 3 | 1 | 0.056 | 0.12 | 0.024 | 0.24 | $1.80 \times 10^{-11}$ | 48 | $6.00 \times 10^{-11}$ | 4537 | 3485 | 4949 |
| 4 | 1 | 0.07 | 0.08 | 0.016 | 0.2 | $1.50 \times 10^{-11}$ | 48 | $6.00 \times 10^{-11}$ | 2390 | 1852 | 2621 |
| 5 | 1 | 0.07 | 0.1 | 0.02 | 0.24 | $1.80 \times 10^{-11}$ | 32 | $4.00 \times 10^{-11}$ | 884 | 714 | 952 |
| 6 | 1 | 0.07 | 0.12 | 0.024 | 0.16 | $1.20 \times 10^{-11}$ | 40 | $5.00 \times 10^{-11}$ | 3493 | 2590 | 3718 |
| 7 | 1 | 0.084 | 0.08 | 0.02 | 0.16 | $1.80 \times 10^{-11}$ | 40 | $6.00 \times 10^{-11}$ | 758 | 557 | 791 |
| 8 | 1 | 0.084 | 0.1 | 0.024 | 0.2 | $1.20 \times 10^{-11}$ | 48 | $4.00 \times 10^{-11}$ | 1661 | 1132 | 1727 |
| 9 | 1 | 0.084 | 0.12 | 0.016 | 0.24 | $1.50 \times 10^{-11}$ | 32 | $5.00 \times 10^{-11}$ | 885 | 852 | 986 |
| 10 | 2 | 0.056 | 0.08 | 0.024 | 0.24 | $1.50 \times 10^{-11}$ | 40 | $4.00 \times 10^{-11}$ | 2013 | 1347 | 2080 |
| 11 | 2 | 0.056 | 0.1 | 0.016 | 0.16 | $1.80 \times 10^{-11}$ | 48 | $5.00 \times 10^{-11}$ | 3760 | 2819 | 3995 |
| 12 | 2 | 0.056 | 0.12 | 0.02 | 0.2 | $1.20 \times 10^{-11}$ | 32 | $6.00 \times 10^{-11}$ | 4702 | 3585 | 5169 |
| 13 | 2 | 0.07 | 0.08 | 0.02 | 0.24 | $1.20 \times 10^{-11}$ | 48 | $5.00 \times 10^{-11}$ | 2463 | 1774 | 2463 |
| 14 | 2 | 0.07 | 0.1 | 0.024 | 0.16 | $1.50 \times 10^{-11}$ | 32 | $6.00 \times 10^{-11}$ | 1553 | 1083 | 1620 |
| 15 | 2 | 0.07 | 0.12 | 0.016 | 0.2 | $1.80 \times 10^{-11}$ | 40 | $4.00 \times 10^{-11}$ | 1624 | 1350 | 1761 |
| 16 | 2 | 0.084 | 0.08 | 0.024 | 0.2 | $1.80 \times 10^{-11}$ | 32 | $5.00 \times 10^{-11}$ | 449 | 321 | 449 |
| 17 | 2 | 0.084 | 0.1 | 0.016 | 0.24 | $1.20 \times 10^{-11}$ | 40 | $6.00 \times 10^{-11}$ | 1646 | 1510 | 1782 |
| 18 | 2 | 0.084 | 0.12 | 0.02 | 0.16 | $1.50 \times 10^{-11}$ | 48 | $4.00 \times 10^{-11}$ | 1680 | 1271 | 1816 |

**Table 9.** Results of range analysis of factors affecting hyporheic exchange at observation point I.

| Factors | $u$ | $H$ | $h$ | $L$ | $\kappa$ | $\theta$ | $D_m$ |
|---|---|---|---|---|---|---|---|
| $K_1$ | 1361.3 | −428.5 | −56.5 | 100.0 | 553.5 | −361.8 | −463.7 |
| $K_2$ | −236.7 | −87.2 | 76.7 | 133.2 | −251.0 | −82.2 | 170.5 |
| $K_3$ | −1124.7 | 515.7 | −20.2 | −233.2 | −302.5 | 444.0 | 293.2 |
| $R_j$ | 2486.0 | 944.2 | 133.2 | 366.3 | 856.0 | 805.8 | 756.8 |
| Susceptibility | | | $u > H > \kappa > \theta > D_m > L > h$ | | | | |

**Table 10.** Results of range analysis of factors affecting hyporheic exchange at observation point II.

| Factors | $u$ | $H$ | $h$ | $L$ | $\kappa$ | $\theta$ | $D_m$ |
|---|---|---|---|---|---|---|---|
| $K_1$ | 962.7 | −397.5 | 24.5 | 14.0 | 392.5 | −280.2 | −403.7 |
| $K_2$ | −171.3 | −59.5 | 47.7 | 104.2 | −201.7 | −43.5 | 123.5 |
| $K_3$ | −791.3 | 457.0 | −72.2 | −118.2 | −190.8 | 323.7 | 280.2 |
| $R_j$ | 1754.0 | 854.5 | 119.8 | 222.3 | 594.2 | 603.8 | 683.8 |
| Susceptibility | | | $u > H > D_m > \theta > \kappa > L > h$ | | | | |

**Table 11.** Results of range analysis of factors affecting hyporheic exchange at observation point III.

| Factors | $u$ | $H$ | $h$ | $L$ | $\kappa$ | $\theta$ | $D_m$ |
|---|---|---|---|---|---|---|---|
| $K_1$ | 1476.4 | −492.9 | −36.1 | 96.4 | 582.9 | −364.2 | −504.2 |
| $K_2$ | −272.9 | −111.6 | 74.3 | 163.6 | −270.4 | −102.2 | 144.3 |
| $K_3$ | −1203.6 | 604.4 | −38.2 | −260.1 | −312.6 | 466.4 | 359.9 |
| $R_j$ | 2680.0 | 1097.3 | 112.5 | 423.7 | 895.5 | 830.7 | 864.2 |
| Susceptibility | | | $u > H > D_m > \theta > \kappa > L > h$ | | | | |

Figure 5 shows a histogram of range analysis of $t$ for CaCl$_2$ concentration at the observation points. As shown in Figure 17, $u$ and $H$ (especially $\kappa$) have dominant effects on hyporheic exchange; $k$, $D_m$, and $\theta$ also have significant effects on hyporheic exchange, whereas $L$ and $h$ ($L > h$) have relatively low effects. These results are consistent with those of previous studies [46,64]. For instance, Wörman et al. [64] reported a dominant effect of surface water flow velocity on hyporheic exchange and a significant effect of $H$.

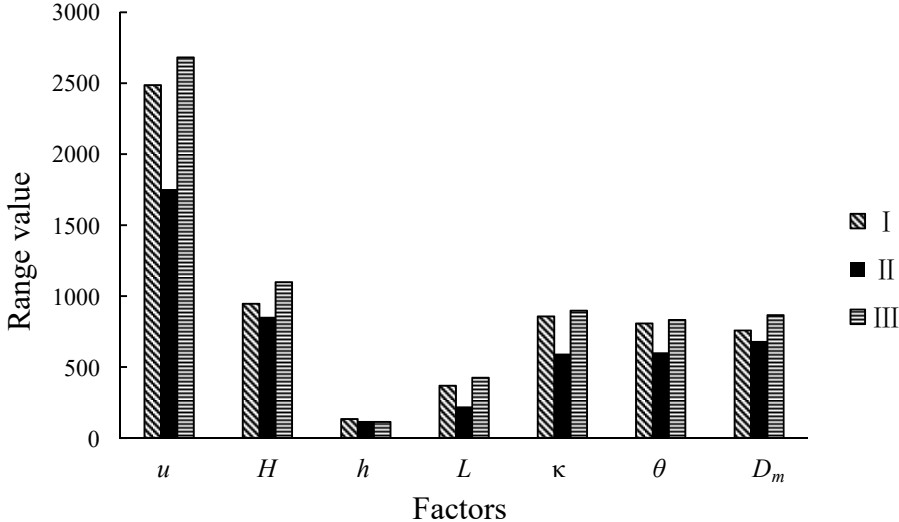

**Figure 17.** Ranges of factors affecting hyporheic exchange at observation points I, II, and III.

*3.7. Migration Routes of Solutes*

Cloud charts of stress distribution at the 6th dune and solute migration routes at different moments in different schemes were obtained based on the sensitivities of factors affecting the hyporheic exchange. Figure 18 shows the cloud chart of stress distribution at the 6th dune in Test 16 (red arrows denote the pore seepage field) and Figure 19 shows solute migration routes at 0 min, 60 min, 180 min, 360 min, 540 min, and 720 min. The maximum pressure was observed at half of the upstream face, whereas the downstream face and the trough were negative pressure zones. As the pressure difference causes exchange of surface water and subsurface water, the pore seepage field was divided into two parts. On the upstream face, reverse flows were observed at locations with heights lower than that corresponding to the maximum pressure, whereas normal flows were observed at other locations. Meanwhile, solute fields at the three observation points approached the surface water until disappearance, as shown in Figure 19.

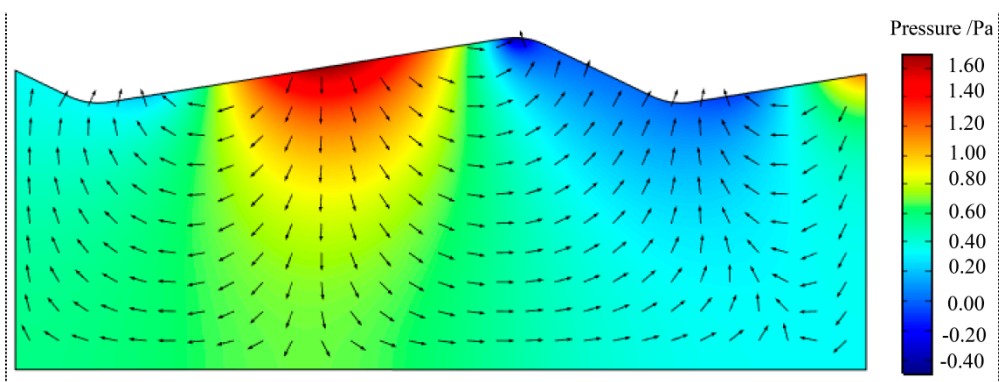

**Figure 18.** Cloud chart of pressures and distribution of seepage field at the 6th dune in Test 16.

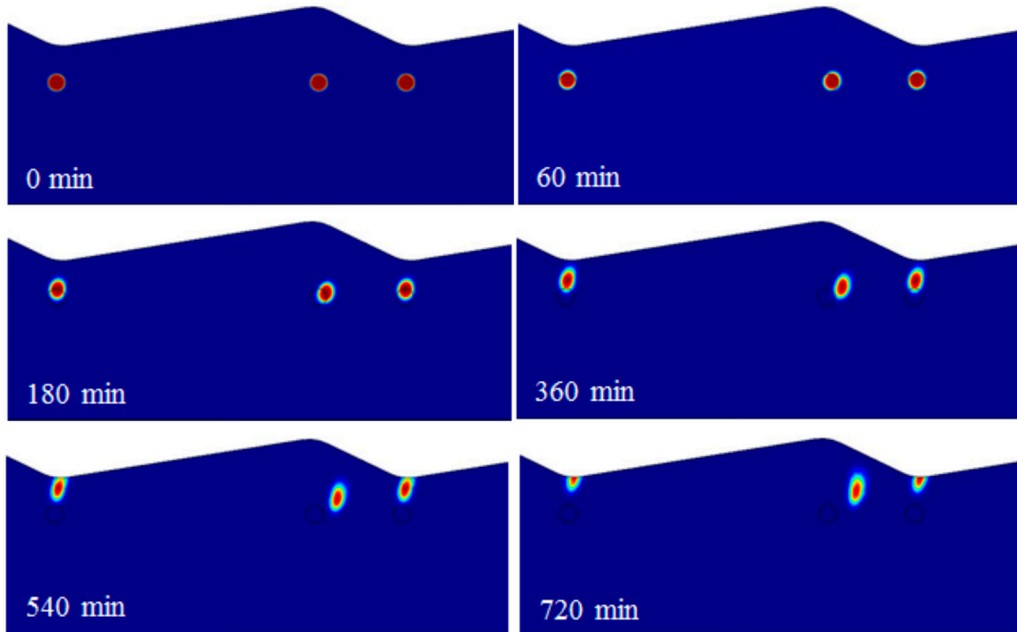

**Figure 19.** Migration routes of solutes at observation points I, II, and III at the 6th dune in Test 16.

## 4. Conclusions

A 2D dune-shaped surface−subsurface coupling mathematical model based on the RANS equation, the $k-\omega$ turbulence model, and a steady state groundwater flow model was proposed. Janseen et al. [55] used exactly the same combination of the multiphysics computational fluid dynamic (CFD) approach and COMSOL Multiphysics to solve these equations. The coupling mathematical model is verified by previous flume experiments [55], and numerical model simulations were verified against observations using the RMSE, $R^2$, and Re as evaluation indices. The maximum RMSE obtained was 0.0055 m s$^{-1}$, significantly lower than the average flow velocity; $R^2 > 0.9$ in all cases, indicating good fitting effectiveness; Re was 3.26–8.53%. In summary, simulation results were highly consistent with experimental results; therefore, it is argued that the proposed calculation model is reliable.

The single-factor effect of $u$, $H$, $h$, and $\kappa$ on hyporheic exchange were investigated by numerical simulations. The results indicated that $u$, $h$, and $\kappa$ were positively related to hyporheic exchange, whereas $H$ was negatively related to hyporheic exchange.

Sensitivity analysis of parameters in the surface−subsurface coupling model and the solute migration model was conducted using orthogonal tests based on the simulation of dye injection into the dune with $t$ used as the evaluation parameter. The results indicated that the sensitivities of $u$ and $H$ were the highest, followed by those of $\kappa$, $D_\mathrm{m}$, $\theta$, $L$, and $h$ ($L > h$).

Solute migration routes at different moments in different schemes were then obtained. Owing to the exchange of surface water and subsurface water, solute fields at monitoring points A, C, and D approached the surface water until disappearance, whereas the solute field at monitoring point B approached the bed substrate.

The hyporheic exchange was shown to be affected by interactions of multiple factors and $u$ and $H$ exhibited the largest effects on hyporheic exchange. The present study provides knowledge vital to the protection and recovery of riverine ecology.

**Author Contributions:** J.R. and X.W. jointly analyzed the data and wrote the paper. Y.Z., B.C., L.M. provided critical feedback on the manuscript.

**Funding:** This work was supported by the National Natural Science Foundation of China (Grant No. 51679194, 51579014).

**Conflicts of Interest:** The authors declare no conflict of interest.

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
