# Peer review of "An Analysis of the Factors Affecting Hyporheic Exchange based on Numerical Modeling"

_water, doi:10.3390/w11040665_

Round 1

Reviewer 1 Report

The reviewer wants to thank the authors for their paper about the numerical simulation of the hyporheic exchange zone based on a previous experimental investigation. He/she has to mention that she/he is not an expert in this specific research field but hopes to make the paper readable for a wide audience. The following points/questions should be addressed:

-      Web of Science delivers no result for “HyporheicExchange”. The reviewer is assuming that the authors wanted to include a space character in between. There are further missing space characters in the text, especially in connections with variables. Please check the complete paper again for such a mistake.  

-      Line (L) 14 / abstract: Based on the abstract the reviewer expected to read about a new combined model, which was checked with the combination of a two commercial codes: “… model was verified by using … ”Please clarify this in the text. 

-      L65: Tonina and Buffington [48] missing

-      L84: studies and blanks before [53] 

-      L79-90: The reviewer was a little bit surprised that Janssen et al. [53] not only conducted the experiments but also already have use the Fluent-Comsol-combination in their paper. This reduces the novelty dramatically to a parameter study and the test of the reproducibility of the previous results. Is this correct? The novelty of the paper should be stated very clearly, especially in this case. 

-      Section 2.1 and 2.2: the section should not simply repeat information, which is available in a text book or the manual of the software. It would be far more important to clearly summarise the assumption of each model and a clear description; which parameter are exchanged! 

-      L95/96: The sentence is misleading! The turbulence is not simulated with a RANS-model. A turbulence model has to be included to solve the RANS-equation. 

-      L117: are those the standard values or are they changed? In both cases WHY shown/changed?

-      L122 equation (9) the variable k has to be checked. 

-      L160: The reviewer first wanted to ask why those values were chosen but it is based on the value of [53]. Please include this. 

-      Figure 1 /L180/ general comment: Please clarify the used velocity boundary condition (profile or one a constant velocity). Same thing for the pressure outlet? Which pressure level was used? L183 is there a difference in the definition of the ce to ef? Please clarify which assumptions are conducted by choosing those boundary conditions. 

-      Fig.2 should include the surface points of Fig. 3 or at least give a reference to the other figure. 

-      L 175 Section 2.4: Did the authors conduct a grid test? Are the results independent of the mesh? 

-      Table 2 has two header rows. 

-      L227 3.1 only verify the surface water part. And the monitoring points should be introduced.

-      Figure 3: Reference is missing and how did the authors get the full data set? Was it provided by the authors of the original paper? 

-      Table 8: Which combinations are new? 18 runs were conducted to evaluate the three parameters. Is this correct or did the reviewer miss something? 

The reviewer is looking forward to read the corrected version of the paper ones again in detail. Thank you! 

Author Response

We appreciate the time and efforts by the editor and reviewer in reviewing this manuscript. The comments provided were all valuable and very helpful for revising and improving our paper. We have made modifications which we hope meet publication requirements of ‘Water'. A revised manuscript with all changes clearly marked in the "Track Changes" function was uploaded and the revised parts were clearly explained in this response. We hope that this response can help the editor and reviewer compare the original and the revised manuscripts and understand why changes were made.

Reviewer 2 Report

The paper should be carefully checked and technically corrected.

Units in the manuscript should be uniformed.

Author Response

(The authors gave the same response as above.)

Round 2

Reviewer 1 Report

The reviewer wants to thank the authors for their corrections. He/she would recommend the following changes: 

-      New line 94 please clary state that Janseen et al. [53] exactly used those combination of software. Please also repeat this comment in the summary, so it is at least well documented.

-      If standard values are used, this chapter could be shortened significantly.

-      Validation is not the same as a verification and in this specific case a grid test. This should be done for every numerical simulation and using only one grid is not good practice. 

Please allow a general comment/feedback: the novelty of the paper is not very high and this comment about the recreation of the data could be seen critical. It would be far better, when the hole paper would be reduced dramatically as well as the parameter study increased. Thank you! 

Author Response

Response to Comments from Reviewer 1

Point 1:

New line 94 please clary state that Janseen et al. [53] exactly used those combination of software. Please also repeat this comment in the summary, so it is at least well documented.

Response 1:

Thanks for the reviewer’s valuable suggestions. We have re-written this two parts to address this comment.

Point 2:

If standard values are used, this chapter could be shortened significantly.

Response 2:

Thanks for the reviewer’s suggestions. We consider keeping this chapter, so that readers understand the principle.

Point 3:

Validation is not the same as a verification and in this specific case a grid test. This should be done for every numerical simulation and using only one grid is not good practice.

Response 3:

Thanks for the reviewer’s suggestions. We have re-written this part to address this comment. 
